# Technological Skills in Higher Education—Different Needs and Different Uses

Ana Luísa Rodrigues [1,*], Luísa Cerdeira [1], Maria de Lourdes Machado-Taylor [2] and Helena Alves [3]

1 Institute of Education, University of Lisbon, 1649-004 Lisboa, Portugal; luisa.cerdeira@ie.ulisboa.pt
2 Center for Research on Higher Education Policies (CIPES), 4450-227 Matosinhos, Portugal; lmachado@cipes.up.pt
3 Department of Management and Economics, University of Beira Interior, 6201-001 Covilhã, Portugal; halves@ubi.pt
* Correspondence: alrodrigues@ie.ulisboa.pt

**Abstract:** Technological skills development is a central issue for a country's educational and social policies. Throughout their school career, from primary to secondary education and later in higher education, students have the opportunity to build and develop various skills, including technological. Considering the different needs and different uses that these skills encompass, these will certainly be necessary and useful in students' academic and professional life. This study reports on an investigation of technological skills development in higher education. It aims to analyze the relevance of technologies integration, which technological skills are built and developed by higher education students, and what their perception about the importance of technological skills is. Based on a literature review, an online questionnaire was designed and applied to 217 students from three public higher education institutions located in the North, Center, and South of Portugal. This intended to verify which areas of technological skills (from the European Digital Competence Framework for Citizens) are most developed and to understand respective repercussions. It is concluded that the balanced development of students' technological skills in higher education is crucial for their personal, social, and professional future and consequently, for their quality of life, with the integration of digital technologies being relevant in the change of the academic work organization, in the relations between learners, teachers, and institutions, and in the new ways of teaching and learning.

**Keywords:** technological skills; higher education; Digital Economy and Society Index (DESI); quality of life





## 1. Introduction

After the COVID-19 pandemic, the need to develop technological competence was evident, and the privileged location for this feat will be throughout school life, from primary and secondary education to higher education. The pandemic has had a severe impact on education, establishing the importance of technological skills and the role that HEIs (Higher Education Institutions) can play in their development by their students.

Especially at this level, it was a quick change from face-to-face lectures to online learning. However, the crisis raises questions about the value offered by a university education, which includes educational content, but also networking and social opportunities. In this way, universities will need to reinvent their learning environments and should promote the development of technological skills in students and teachers [1,2].

According to the Organization of Ibero-American States for Education, Science and Culture (OEI) report 2020, the situation experienced, with the need to use the so-called emergency remote education, gave visibility to an unresolved problem, of using distance learning. It placed HEIs before the need to define priorities in terms of curricular content and to focus educational efforts on the competences and skills that are in fact essential in times of uncertainty. These skills include technological skills, but also soft skills that allow

thinking, and understanding and transfer of knowledge, which contribute to personality development and promote students' intellectual autonomy [3].

Thereby, in the present information and knowledge society, characterized by globalization, changes in the labor market, and the exponential growth of technologies, students face new and different learning needs with the wide dissemination and accessibility of information, as well as the need to develop skills and competences to understand and interpret information to be able to transform it into knowledge.

Technological skills are already essential in today's knowledge society and appear to be crucial to peoples' future life satisfaction, alongside generic skills. It was found that the main skills of the 21st century, critical thinking, problem-solving, communication, and technological skills, as well as age and income, have a positive impact on life satisfaction [4].

Information Communication Technologies (ICTs), especially the Internet and the web, have changed every aspect of human life, from individual social life to industrial modernization and the economic growth of nations. Technology is critical to our wellbeing and above all, it has become a necessity that contributes to expanding the possibilities of having a better Quality of Life (QoL), a fundamental element for the sustainable development of countries and regions.

In a study on the use of ICTs, and European citizens' assessment of their QoL, results reveal a clear relationship between technology and quality of life. It was concluded that life satisfaction increases with greater use of technologies and ICT preparedness. Thus, the digital citizen is happier living in regions with technological capacity, which invest in R&D, and which are committed to achieving sustainable growth [5].

Another study on the role of ICTs on the QoL shows that digital inclusion and ICT access significantly influence QoL at the global level and reveals that the use of ICT is significant in the linkage between digital inclusion and QoL, but not for the linkage between ICT access and QoL [6]. Consequently, in developing QoL policies, policymakers should provide ICT access, creating a digitally enabled society with access to affordable ICT that will enable people to enhance their skills, avoiding social exclusion, and promote their QoL.

This study allows exploring the knowledge and reflecting on the importance of digital technologies in higher education, especially at the level of the balanced development of the various areas of students' technological competences and their possible repercussions, both of the students and of their influence on pedagogical and organizational processes. Therefore, this study aims to analyze which technological skills are being built and developed by higher education students and what their perception about the importance of this skills in their professional life and quality of life is.

There are several studies on the importance of skills and the use of technologies, methodologies, and teaching and learning with technologies in higher education. However, studies on the needs and uses of technology skills development by higher education students and focusing on the relationship between technology and quality of life are few. Thus, it is intended to demonstrate the relevance of promoting technological skills development in higher education students.

## 2. The Relevance of Technologies in Higher Education and Society

The integration of digital technologies in higher education in the teaching and learning processes is unquestionable today [2,7].

Digital innovation can provide economic growth and generate value for business and benefits for society, for instance, by creating jobs, reducing inequality, and promoting inclusiveness. At the same time, it can boost the achievement of the UN (United Nations) Sustainable Development Goals and support its three pillars: improving people's quality of life, fostering equitable growth, and protecting the environment, according to the Digital Transformation Initiative. On the other hand, at the education level, personalized and automated learning services will help to deliver a personalized and individual approach based on unique needs, being able to monitor what a student has learned more effectively [8].

We live in an increasingly complex digital world where it is intended that students behave as active citizens. They need to develop knowledge and skills to make the most of the digital revolution, for example, they must be able to fill out an online job application, use e-commerce to make purchases, or make bank transactions through an application. The lack of technological skills can have a profound effect on people's overall life opportunities and employability. In order to assess the potential of digital technologies, the European Commission developed the European Digital Competence Framework for Citizens, known as DigComp. This identifies 21 competencies in five main areas, describing what it means to have digital knowledge, and it is necessary to have competencies in each of these areas to achieve objectives related to work, employability, learning, leisure, and participation in society [9].

Being digitally competent is about being able to use such digital technologies in a critical, collaborative, and creative way. The competence areas defined are as follows: (1) information and data literacy competence; (2) communication and collaboration competence; (3) digital content creation competence; (4) safety competence, and (5) problem-solving. Eight proficiency levels for each competence have been defined through four learning outcomes, each with two levels: foundation, intermediate, advanced, highly specialized [9].

The Digital Competence Framework can help to monitor citizen's technological skills and to define public policies, allowing knowing the digital competence at the country level. Using the DigComp framework, the EU-wide Digital Economy and Society Index was created, allowing measuring of human capital, which is needed to take advantage of the possibilities offered by a digital society.

The Digital Economy and Society Index (DESI) is a composite index that summarizes relevant indicators on Europe's digital performance and tracks the evolution of EU Member States, calculated as the weighted average of five main dimensions: (1) connectivity (25%), (2) human capital (25%), (3) use of Internet (15%), (4) integration of digital technology (20%), and (5) digital public services (15%) (see Figure 1) (official website of the European Union, https://ec.europa.eu/jrc/en/digcomp, accessed on 10 February 2021).

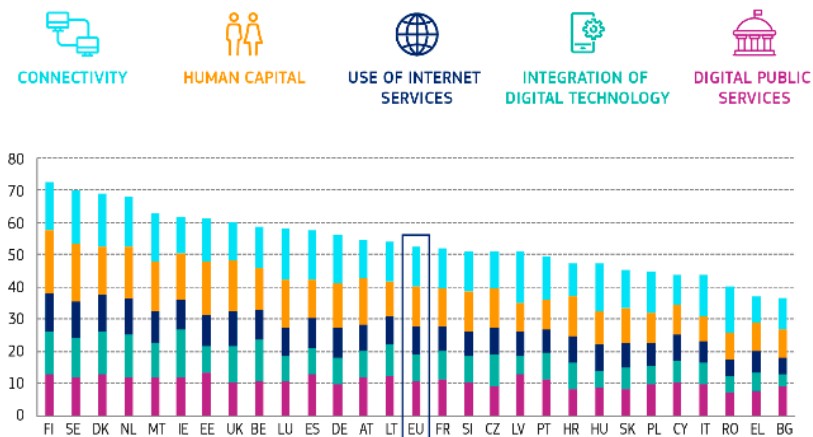

**Figure 1.** The Digital Economy and Society Index (DESI), 2020. Source: official website of the European Union, accessed on 10 February 2021.

In 2019, all EU countries improved their digital performance. The countries that obtained the best classification in DESI 2020 were Finland, Sweden, Denmark, and the Netherlands, which are among the global leaders in digitalization. These countries were followed by Malta, Ireland, and Estonia. Portugal was in 19th position, below the EU average [10].

Analyzing the Portugal DESI, we can verify that in 2020, compared to the previous year, progress was being made in the dimension of human capital, due to an improvement in the basic level of technological skills and a higher proportion of ICT graduates. This issue is particularly significant for Portugal, given the current low level of digital literacy in

its population. However, the country continued to underperform by European standards in human capital and use of Internet services (Figure 2) [10].

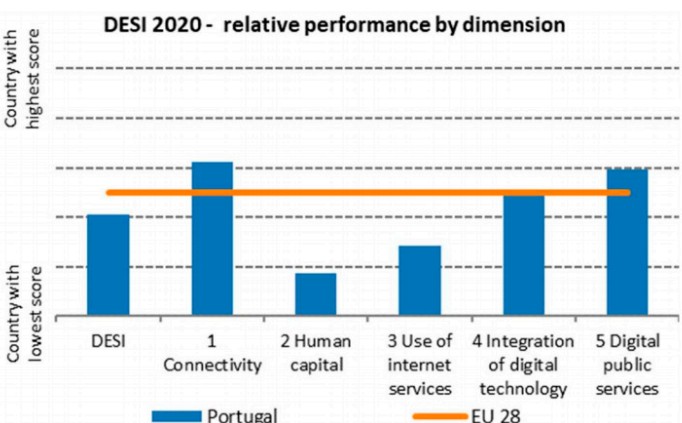

**Figure 2.** DESI 2020 by dimension in Portugal (accessed on 10 February 2021) [10].

Regarding the dimension of connectivity, Portugal fell one place, compared to the previous year, but its overall score is above average. This is mainly due to the significant results in the deployment of very high capacity networks, as well as broadband connections of at least 100 Mbps. The indicator that recorded the biggest drop is found in digital technology in the business area, standing in 2020 below the EU average. In digital public services, Portugal continued to perform well.

Portugal continues to implement the national technological skills initiative INCoDe.2030 and launched the second phase of a national strategy for digitizing the economy and two relevant strategies in Artificial Intelligence and advanced computing, strongly focused on improving advanced technological skills.

The current crisis arising from the Covid-19 pandemic is having a significant impact on the main societal indicators related to the use of Internet services by citizens. As Europe progressively exits the pandemic, the recuperation must be planned with special attention. To face this crisis, Portugal adopted a range of measures in the digital domain, including the development of platforms and applications in health and education, the promotion of public administration digitization, and the offer of new services. Several initiatives provided support for digital teaching at home, and national and regional telescope channels were created, with the aim of helping students without access to the Internet or adequate equipment.

Looking to the future, Portugal is at an advanced stage of network deployment and is above the EU average in terms of the provision of digital public services. On the other hand, it is late in the transition to 5G and has a poor performance in the indicators related to technological skills [10].

## 3. Methods

This paper is based on a literature review, a descriptive analysis of secondary statistical data, and the analysis of primary data collected through an online questionnaire to students of HEIs.

The construction of the questionnaire was based on the areas of competence of the European Digital Competence Framework for Citizens [9].

This aims to analyze (i) how Digital Technologies (DT) are used by higher education students in Portugal, at a personal/social and academic/professional level; (ii) what technological skills students develop throughout their academic career, and (iii) what the impact/relevance of DT on their professional future and quality of life is. In the process of questionnaire elaboration, the validity and reliability were ensured by being very rigorous and clear regarding the collected data, the explicit choice of sample selection, the use of standardized recording methods, the testing of the instrument, and the care

in the appropriate statistical treatment of data. In addition, the four authors of the study participated in questionnaire design and test, thus allowing the use of the triangulation technique with the participation and analysis by different researchers [11–13].

The questionnaire was tested with four students, one from each level of education. The answers were analyzed in order to understand any imprecision, incongruity, or lack of answer or understanding of the questions, with some reformulations being made. Subsequently, it was reviewed by three experts, experienced researchers, who were asked to give their feedback on correctness of form, objectivity/subjectivity, redundancy, and adequacy to the research questions and objectives, and suggestions for improvement, which allowed the refinement of the questionnaire.

Posteriorly, it was applied anonymously and distributed to all students enrolled in two faculties of the universities of ( ... ), and a school of ( ... ) polytechnic, located in the North, Center, and South of Portugal, between November 2020 and January 2021 [11]. In 2020, 82% of Portuguese students studied in public higher education, of which 62% were in university education and 38% were in polytechnic [14]. Therefore, we chose to apply the questionnaire only in these public institutions. It is a convenience sample and therefore not representative of the population.

In terms of structure, the questionnaire was divided into four parts: personal data, use of DT, development of technological skills, and impact of DT. It included thirteen closed-answer questions and three open-answer questions. A total of 217 valid responses were received. The total number of students enrolled in public higher education in Portugal in 2020 was 108,671, so the responses correspond to 0.2% of this population [14].

The data analysis was carried out based on descriptive statistics, and the open questions were analyzed using the content analysis technique [15,16], with the support of the NVivo software.

## 4. Results

The results obtained from the literature review and questionnaire analysis allow a better understanding of the importance of the development of technological skills and leave some clues about the students' perception regarding their influence on their professional future and quality of life.

Of the 217 students who answered the questionnaire, 71% were female and 29% were male. The majority, 44%, were under 24 years old, however 29% of respondents were 40 years old or more. Regarding the level of education attended, 42% were in undergraduate, 36% in master/graduate, and 18% in doctorate (view Figure 3).

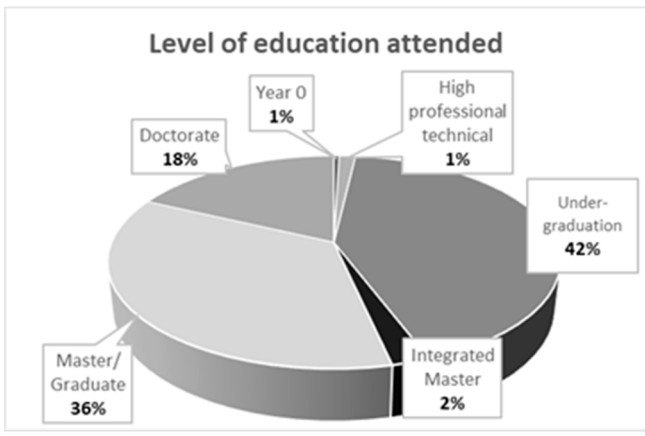

**Figure 3.** Level of education attended.

Considering the main digital equipment used, respondents mainly used, every day, the smartphone (82% + 6%) and the portable computer (59% + 20%). The desktop computer, tablet, and simple mobile phone were never or less than once a week used by 70%, 68%, and 78%, respectively (Table 1).



**Table 1.** Frequency of digital equipment use.

| | Desktop Computer | | Portable Computer | | Tablet | | Smartphone | | Mobile Phone | |
|---|---|---|---|---|---|---|---|---|---|---|
| | *n* | % | *n* | % | *n* | % | *n* | % | *n* | % |
| Never or less than once a week | 151 | 70% | 6 | 3% | 147 | 68% | 9 | 4% | 170 | 78% |
| A few times a week | 10 | 5% | 37 | 17% | 32 | 15% | 18 | 8% | 11 | 5% |
| Once a week | 6 | 3% | 2 | 1% | 12 | 6% | 0 | 0% | 5 | 2% |
| Every day, 1 or 2 times | 13 | 6% | 43 | 20% | 9 | 4% | 12 | 6% | 4 | 2% |
| Every day, several times | 37 | 17% | 129 | 59% | 17 | 8% | 178 | 82% | 27 | 12% |

The most usually used applications and software signalized were text editor (e.g., Word) with 94%, WhatsApp with 93%, learning platform (e.g., Moodle, Classroom, and Teams) with 87%, presentations (e.g., PowerPoint) with 84%, and Instagram with 73% (Table 2).

**Table 2.** Applications/software usually used.

| | *n* | % |
|---|---|---|
| Text editor (e.g., Word) | 204 | 94% |
| Spreadsheet (e.g., Excel) | 128 | 59% |
| Presentations (e.g., PowerPoint) | 182 | 84% |
| Learning platform (e.g., Moodle, Classroom, and Teams) | 189 | 87% |
| Questionnaires (e.g., Google Forms and Kahoot) | 110 | 51% |
| Data-processing programs (e.g., SPSS and NVivo) | 52 | 24% |
| Bibliographic references (e.g., EndNote and Mendeley) | 58 | 27% |
| Facebook | 155 | 71% |
| WhatsApp | 202 | 93% |
| Instagram | 159 | 73% |
| LinkedIn | 65 | 30% |
| Twitter | 11 | 5% |

In terms of frequency by type of digital technologies used, most respondents used them every day, in their personal/social lives (74% + 15%), academic/professional activities (54% + 27%), and teaching/learning activities (47% + 27%). The telecommuting was never or less than once a week used for 43% of respondents, 17% used it a few times a week, and 21% + 10% used it every day (Table 3).

**Table 3.** Frequency by type of use of digital technologies.

| | Personal/Social | | Academic/ Professional | | Teaching/ Learning | | Telecommuting | |
|---|---|---|---|---|---|---|---|---|
| | *n* | % | *n* | % | *n* | % | *n* | % |
| Never or less than once a week | 7 | 3% | 7 | 3% | 3 | 1% | 94 | 43% |
| A few times a week | 8 | 4% | 25 | 12% | 39 | 18% | 37 | 17% |
| Once a week | 9 | 4% | 10 | 5% | 15 | 7% | 18 | 8% |
| Every day, 1 or 2 times | 33 | 15% | 58 | 27% | 58 | 27% | 22 | 10% |
| Every day, several times | 160 | 74% | 117 | 54% | 102 | 47% | 46 | 21% |

Among the most marked activities developed with digital technologies were search engine navigation and searches (e.g., Google) with 97%, send messages and/or use email with 96%, perform academic work with 89%, use of social networks (Facebook, Twitter, Instagram, etc.) with 88%, use of learning support platforms (e.g., Moodle, Classroom, and Teams), and chat and/or videoconference, both with 87% (Table 4).

**Table 4.** Activities developed with digital technologies.

|  | *n* | *%* |
|---|---|---|
| Search engine navigation and searches (e.g., Google) | 210 | 97% |
| Watching movies/videos and/or listening to music | 178 | 82% |
| Chat and/or videoconference | 189 | 87% |
| Send messages and/or use email | 208 | 96% |
| Online shopping | 129 | 59% |
| Use social networks (Facebook, Tweeter, Instagram, etc.) | 191 | 88% |
| Use academic/professional social networks (LinkedIn, ResearchGate, etc.) | 98 | 45% |
| Attend classes and/or videoconferences | 180 | 83% |
| Perform academic work | 194 | 89% |
| Use learning support platforms (e.g., Moodle, Classroom, and Teams) | 188 | 87% |
| Consult databases and bibliography | 152 | 70% |
| Perform professional tasks | 102 | 47% |

Considering the competence areas from European Digital Competence Framework for Citizens [9], respondents were asked about the degree of technological skills developed during higher education. With the highest percentage of "good" responses, the following technological skills were recorded: "Adapt my searching strategy to find the most appropriate information and content in digital environments", "Manipulate information for easier organization, storage and retrieval", and "Use and select appropriate digital technologies to interact". With a "very good" response, "Adapt my searching strategy to find the most appropriate information and content in digital environments" and "Share information and content using a variety of digital tools" also stand out. The technological skills referred to as "weak" developed were "Apply rules of copyright and licenses", "Choosing solutions to protect the environment from the impact of digital technologies", and "Solve problem situations in digital environments" (see Table 5).

**Table 5.** Technological skills developed during higher education.

| Technological Skills | Very Weak/None | | Weak | | Moderate | | Good | | Very Good | |
|---|---|---|---|---|---|---|---|---|---|---|
| | *n* | *%* | *n* | *%* | *n* | *%* | *n* | *%* | *n* | *%* |
| Carry out an evaluation of the credibility and reliability of different sources of data and information | 7 | 3% | 20 | 9% | 68 | 31% | 88 | 41% | 34 | 16% |
| Adapt my searching strategy to find the most appropriate information and content in digital environments | 4 | 2% | 7 | 3% | 62 | 29% | 99 | 46% | 45 | 21% |
| Manipulate information for easier organization, storage, and retrieval | 5 | 2% | 16 | 7% | 60 | 28% | 100 | 46% | 36 | 17% |
| Use and select appropriate digital technologies to interact | 4 | 2% | 13 | 6% | 67 | 31% | 99 | 46% | 34 | 16% |
| Share information and content using a variety of digital tools | 4 | 2% | 11 | 5% | 74 | 34% | 85 | 39% | 43 | 20% |
| Apply referencing and attribution practices | 8 | 4% | 31 | 14% | 67 | 31% | 76 | 35% | 35 | 16% |
| Propose different digital services to participate in society as a citizen | 12 | 6% | 35 | 16% | 85 | 39% | 67 | 31% | 18 | 8% |
| Select digital tools and technologies for collaborative processes | 9 | 4% | 34 | 16% | 67 | 31% | 80 | 37% | 27 | 12% |
| Use communication strategies in digital environments to an audience | 11 | 5% | 38 | 18% | 74 | 34% | 74 | 34% | 20 | 9% |
| Create and edit content in different formats | 12 | 6% | 35 | 16% | 67 | 31% | 66 | 30% | 37 | 17% |

**Table 5.** *Cont.*

| Technological Skills | Very Weak/None | | Weak | | Moderate | | Good | | Very Good | |
|---|---|---|---|---|---|---|---|---|---|---|
| | *n* | % | *n* | % | *n* | % | *n* | % | *n* | % |
| Apply rules of copyright and licenses | 17 | 8% | 41 | 19% | 68 | 31% | 68 | 31% | 23 | 11% |
| Use safety and security measures | 19 | 9% | 37 | 17% | 79 | 36% | 53 | 24% | 29 | 13% |
| Choosing solutions to protect the environment from the impact of digital technologies | 19 | 9% | 42 | 19% | 83 | 38% | 51 | 24% | 22 | 10% |
| Select the most appropriate digital tools and technological responses | 10 | 5% | 29 | 13% | 75 | 35% | 76 | 35% | 27 | 12% |
| Adjust and customize digital environments to personal needs | 11 | 5% | 29 | 13% | 72 | 33% | 82 | 38% | 23 | 11% |
| Adapt technologies to create knowledge and innovate processes and products | 20 | 9% | 29 | 13% | 79 | 36% | 62 | 29% | 27 | 12% |
| Solve problem situations in digital environments | 16 | 7% | 41 | 19% | 84 | 39% | 56 | 26% | 20 | 9% |
| Recognize how to improve or update my own digital competence needs | 16 | 7% | 31 | 14% | 67 | 31% | 74 | 34% | 29 | 13% |

When asked about the degree of importance, at a professional level in their future, of the skills listed, the respondents referred in a higher percentage with "very important" "Search and select online credible information and content" (66%) and "Use online security and privacy rules" (63%). With "little importance" were more marked "Make presentations online" (6%) and "Try new digital tools" (5%), according to Table 6.

**Table 6.** Skills importance in professional future.

| Skills | Nothing | | Little | | Moderately | | Important | | Very Important | |
|---|---|---|---|---|---|---|---|---|---|---|
| | *n* | % | *n* | % | *n* | % | *n* | % | *n* | % |
| Obtain and mobilize information through digital media | 1 | 0% | 3 | 1% | 21 | 10% | 73 | 34% | 119 | 55% |
| Search and select online credible information and content | 0 | 0% | 5 | 2% | 16 | 7% | 53 | 24% | 143 | 66% |
| Organize and control information in digital format | 0 | 0% | 5 | 2% | 25 | 12% | 68 | 31% | 119 | 55% |
| Communicate and interact with digital technologies | 1 | 0% | 5 | 2% | 18 | 8% | 58 | 27% | 135 | 62% |
| Sharing information and content in online format | 1 | 0% | 5 | 2% | 24 | 11% | 55 | 25% | 132 | 61% |
| Work collaboratively with the use of digital technologies | 1 | 0% | 3 | 1% | 19 | 9% | 60 | 28% | 134 | 62% |
| Make presentations online | 0 | 0% | 12 | 6% | 27 | 12% | 64 | 29% | 114 | 53% |
| Participate in videoconferences | 0 | 0% | 7 | 3% | 28 | 13% | 59 | 27% | 123 | 57% |
| Use online security and privacy rules | 2 | 1% | 3 | 1% | 19 | 9% | 57 | 26% | 136 | 63% |
| Try new digital tools | 0 | 0% | 11 | 5% | 16 | 7% | 89 | 41% | 101 | 47% |
| Create new knowledge and innovate | 0 | 0% | 8 | 4% | 22 | 10% | 71 | 33% | 116 | 53% |
| Improve time management | 0 | 0% | 7 | 3% | 15 | 7% | 63 | 29% | 132 | 61% |
| Achieve greater work productivity | 0 | 0% | 8 | 4% | 16 | 7% | 61 | 28% | 132 | 61% |
| Solve problem situations with the use of technology | 0 | 0% | 8 | 4% | 23 | 11% | 73 | 34% | 113 | 52% |

Regarding the main difficulties in terms of the use and integration of digital technologies, students referred above all to their lack of computer knowledge and the need for support and training. With some references, questions arose about the difficulty of accessing hardware, the need to use different applications and software, and quick and constant updates. The lack of time and excessive use of digital technologies was also mentioned. Four students said they had no difficulties.

In response to the question about the importance of technological skills at a professional level in the future, most students agreed and referred to its inevitability, especially considering its relevance in society and digital world in the future, its demand in the labor market, and the usefulness of technologies. This was mentioned in relation to the organization, time management, dissemination and promotion of knowledge, problem-solving, creativity, and development of recreational activities.

Other references were made on technological skills, such as allowing the increase in efficiency, greater connectivity and communication, performance improvement, information management, work flexibility, time optimization, distance work, and adaptation to a constantly changing world. There was only one reference that devalued the technologies because they were imposed on us.

On the relevance and impact on a personal level of technological skills for their quality of life, students reported that they are essential for accessing knowledge and personal development at various levels, such as for time management, communication, teaching and learning, ease and speed in performing tasks, problem-solving, safety, leisure, and well-being.

However, both advantages and disadvantages of integrating technologies for quality of life were highlighted. Pointing as positive aspects were the optimization of time and resources, the ease of communication and task management, and the possibility of teleworking. Several negative aspects were also mentioned, namely, the dependence on technologies, dispersion of attention, difficulties in managing time between family and work, decreased human interaction, and other issues related to emotional and physical health due to the excess of time spent in front of screens.

## 5. Discussion and Conclusions

Authors should discuss the results and how they can be interpreted from the perspective of previous studies and of the working hypotheses. The findings and their implications should be discussed in the broadest context possible. Future research directions may also be highlighted.

The integration of digital technologies in HEIs is fundamental for the development of students' technological skills and for their professional future and consequently and simultaneously, for the increase of their quality of life.

We found that most respondents used smartphone and portable computer, and applications or software used were the text editor, learning platforms, presentations, WhatsApp, and Instagram, in personal, social, academic/professional, and teaching/learning life. The activities with digital technologies most developed were "Search engine navigation and searches", "Send messages and/or use email", "Perform academic work", and "Use social networks".

Regarding the skills they considered having developed more in higher education, they stressed "Adapt my searching strategy to find the most appropriate information and content in digital environments", "Share information and content using a variety of digital tools", "Manipulate information for easier organization, storage and retrieval", and "Use and select appropriate digital technologies to interact". The areas of technological skills considered to be less developed were those related to security as "Apply rules of copyright and licenses", "Use safety and security measures", or yet "Choosing solutions to protect the environment from the impact of digital technologies", and "Solve problem situations in digital environments".

About their perception of the importance of technological skills in employment, personal, and social life in the future, they confirmed the importance of all skills mentioned, and those most referred to as important were "Search and select online credible information and content" and "Use online security and privacy rules". The least valued were "Make presentations online" and "Participate in videoconferences".

In general, students considered technological skills crucial in the digital world and labor market in the future. The technologies are useful at the level of the organization, for time management, dissemination and promotion of knowledge, and problem-solving, and allow the increase in efficiency, greater connectivity and communication, information management, distance work, and adaptation to a constantly changing world.

The main difficulties in terms of the use and integration of digital technologies identified were the need for support and training in computer science, the lack of time, difficulty in accessing hardware, and the need to use different applications and software.

Regarding their relevance in quality of life, technologies present advantages and disadvantages. On the one hand, they enable the optimization of time and resources, the ease of communication and task management, and the possibility of teleworking. On the other hand, technologies can cause dependency, dispersion of attention, difficulties in managing time between family and work, decreased human interaction, and problems at an emotional and physical level due to their overuse.

With regard to the ongoing digital transformation of economies and societies in OECD countries changing the quantity and quality of jobs [7] and the main conclusions of the report about Future of Jobs de 2020 [17], it is highlighted that (i) skills gaps remain large compared to the skills required in jobs that will change over the next five years; (ii) in the absence of proactive efforts, inequality will increase due to the double impact of technology and the pandemic recession; (iii) online learning and training is increasing strongly. Employees are placing larger emphasis on personal development courses, and the unemployed are focused on learning technological skills; (iv) the requalification of employees has economic effects with significant mid-to-long-term dividends, not only for their enterprise but also for the benefit of society more broadly. The digitalization in teaching and learning will likely occupy a much more prominent place as a policy priority because it can maintain access and quality at a lower cost of instruction, raising efficiency [7].

Nowadays, learning with digital technologies in higher education is normal and included in formal learning environments, in which the students are viewed as active participants in the search for knowledge [18].

After the pandemic of COVID-19, many studies have already been carried out on how technologies were used in higher education especially in this period, for example, at the level of curriculum change [19], and considering motivation [20], teaching methods [21,22], their integration in the pedagogical process [23], the learning systems used [24], or the management of HEIs [25]. However, we will see how this experience can be useful in the near future, when we return to the previous normality.

The results of this study are also important for HEIs at the organizational level, insofar as digitalization of teaching and learning begins to change the economics of higher education and the organization of academic work, because this is not simply a change in the mode of educational delivery from face-to-face to online learning or blended learning, but also has the potential to transform higher education, the academic work organization, and the relations between learners, teachers, and their institutions [7].

Hence, it is relevant to know which technological skills are built and developed by students in higher education, the different needs and uses in the different areas of competence, to promote their balanced development in the future in each of the areas and to reflect on new ways of teaching and learning with quality in higher education.

In order to overcome the limitations of the study, it is important to develop more studies for a deeper understanding of which and how technological skills need to be further promoted, as well as to analyze ways to increase the positive effects of the use

of technologies. It is also relevant that this study can serve as a basis for other research into this subject; by extending the size of the sample to allow generalization at a national level or the possibility of comparison between different countries, further studies can offer sustained information that contributes to the consolidation of knowledge in this area aimed at the scientific community and HEI's for the development of consonant public policies.

**Author Contributions:** Conceptualization, methodology, software, investigation, resources, writing—original draft preparation, A.L.R.; validation, formal analysis, data curation, writing—review and editing, visualization supervision, project administration and funding acquisition, A.L.R.; L.C.; M.d.L.M.-T. and H.A.; All authors have read and agreed to the published version of the manuscript.

**Funding:** This research is part of the U-Value research project funded by Fundação para a Ciência e a Tecnologia (FCT), grant number PTDC/EGE-OGE/29926/2017—The impact of Higher Education Institutions on the quality of life of their regions.

**Informed Consent Statement:** The online questionnaire was applied with informed consent by providing a link to the research project, in an anonymous and confidential format, with the collected data being treated in a global and aggregated way.

**Data Availability Statement:** The data presented in this study are available on request from the corresponding author. The data are not publicly available due to privacy issues. The research on which this study is based follows the general ethical international standards.

**Conflicts of Interest:** The authors declare no conflict of interest. The funders had no role in the design of the study; in the collection, analyses, or interpretation of data; in the writing of the manuscript, or in the decision to publish the results.

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
