# Peer review of "Technological Skills in Higher Education—Different Needs and Different Uses"

_education, doi:10.3390/educsci11070326_

Round 1

Reviewer 1 Report

In the abstract a sentence with the main conclusions must be included;

The sentence below, in the lines 77-78 must be deleted once it is repeated in the lines  159-160 

This paper is based on a literature review, a descriptive analysis of secondary statistical data, and the analysis of primary data collected through an online questionnaire to HEIs students. 

Author Response

Response to Reviewer 1 Comments

This paper is based on a literature review, a descriptive analysis of secondary statistical data, and the analysis of primary data collected through an online questionnaire to HEIs students. 

Point 1: In the abstract a sentence with the main conclusions must be included;

Response 1: Amendment made in the main document.

Point 2: The sentence below, in the lines 77-78 must be deleted once it is repeated in the lines 159-160.

Response 2: Amendment made in the main document.

Reviewer 2 Report

Although the article is interesting, it has points that need to be improved:
- what does this study contribute to the scientific community?
- In relation to the sample, it is very scarce considering the different educational levels and the geographical areas in which it is applied. Therefore, you must increase that sample.
- It must explain the process of elaboration (validity and reliability) of the questionnaire used.
- The references are scarce. Currently in the databases we find a large number of publications on this subject.

Author Response

Response to Reviewer 2 Comments

Although the article is interesting, it has points that need to be improved:

Point 1: what does this study contribute to the scientific community?

Response 1:

This study allows exploring the knowledge and reflecting on the importance of digital technologies in higher education, especially at the level of the balanced development of the various areas of students' technological competences and their possible repercussions, both of the students and their influence on pedagogical and organisational processes. (added in lines 71 to 75)

It is also relevant that this study can serve as a basis for other research into this subject, by extending the size of the sample that allows the generalization at a national level or the possibility of comparison between different countries, in order to offer sustained information that contributes to the consolidation of knowledge in this area aimed at the scientific community and HEI's for the development of consonant public policies. (added in lines 352 to 357)

Point 2: In relation to the sample, it is very scarce considering the different educational levels and the geographical areas in which it is applied. Therefore, you must increase that sample.

Response 2:

This study sought to carry out an exploratory analysis, having opted for a convenience sample with the deliberate choice of only three public institutions. This study did not intend the sample to be representative of the population, focusing only on the dimension of students' perception.

This issue was also identified as one of the limitations of the study. (see lines 371 to 378 of the current document)

Even so, 217 responses were obtained from these three institutions.

The total number of students enrolled in public higher education in Portugal in 2020 was only 108,671, so the responses correspond to 0.2% of the universe. (added in lines 188 and 189)

Point 3: It must explain the process of elaboration (validity and reliability) of the questionnaire used.

Response 3:

The construction of the questionnaire was based on the areas of competence of the European Digital Competence Framework for Citizens.

In the process of questionnaire elaboration, the validity and reliability were ensured by being very rigorous and clear regarding the collected data, the explicit choice of sample selection, the use of standardised recording methods, the testing of the instrument and the care in the appropriate statistical treatment of data. In addition, the four authors of the study participated in questionnaire design and test, thus allowing the use of the triangulation technique with the participation and analysis by different researchers. (added in lines 176 to 181)

The questionnaire was tested with four students, one from each level of education.

The answers were analyzed in order to understand any imprecision, incongruity, lack of answer or understanding of the questions, with some reformulations being made. Subsequently, it was reviewed by three experts, experienced researchers, who were asked to give their feedback on: correctness of form, objectivity/subjectivity, redundancy, adequacy to the research questions and objectives, and suggestions for improvement, which allowed for the refinement of the questionnaire. (added in lines 183 to 188)

Point 4: The references are scarce. Currently in the databases we find a large number of publications on this subject.

Response 4:

References can be considered scarce due to the objective of the article being focused on the analysis of the questionnaire, and a more concise literature review was chosen.

However, we also tried to complement this with the inclusion of some other references (+11) that we considered relevant in this review of the article (see in lines 177 to 182 and in lines 352 to 360).

Round 2

Reviewer 2 Report

Recommendations have been made.
It has to be corrected: where it says "Information Communication Technologies (ICTs)" it has to put "Information Communication Technologies (ICT)"

Author Response

Minor Revisions performed. Revised the paper in blue.

Thank you.
